# Single-Molecule Real-Time Sequencing for Identifying Sexual-Dimorphism-Related Transcriptomes and Genes in the Chinese Soft-Shelled Turtle (*Pelodiscus sinensis*)

**DOI:** 10.3390/ani13233704

**Published:** 2023-11-29

**Authors:** Tong Zhou, Guobin Chen, Jizeng Cao, Jiahui Wang, Guiwei Zou, Hongwei Liang

**Affiliations:** 1Yangtze River Fisheries Research Institute, Chinese Academy of Fisheries Science, Wuhan 430223, China; zhoutong@yfi.ac.cn (T.Z.); cgb1251877642@126.com (G.C.); 15832942914@163.com (J.C.); zougw@yfi.ac.cn (G.Z.); 2College of Fisheries and Life Science, Shanghai Ocean University, Shanghai 201306, China; 3College of Animal Science and Technology, Henan University of Animal Husbandry and Economy, Zhengzhou 450046, China; 18016484699@163.com

**Keywords:** Chinese soft-shelled turtle, *P. sinensis*, sex determination, SMRT sequencing, gene expression

## Abstract

**Simple Summary:**

The Chinese soft-shelled turtle shows obvious sexual dimorphism. The male *P. sinensis* is larger and, thus, more popular in the market. We obtained full-length (FL) transcriptome data and compared the difference between male and female *P. sinensis*. Compared to female individuals, there were 5050 upregulated genes and 3590 downregulated genes in male individuals. Female-biased genes such as *Smad4*, *Wif1*, *Cypj2*, and *17β-hsd* and male-biased genes such as *Nkd2* and *Prp18* were identified by omics and quantitative analysis. The results may facilitate future comparative studies on the transcriptome and gene function related to sex determination in *P. sinensis*.

**Abstract:**

The Chinese soft-shelled turtle (*Pelodiscus sinensis*), an economically important aquatic species in China, displays considerable sexual dimorphism: the male *P. sinensis* is larger and, thus, more popular in the market. In this study, we obtained the full-length (FL) transcriptome data of *P. sinensis* by using Pacific Biosciences (PacBio)’s isoform sequencing and analyzed the transcriptome structure. In total, 1,536,849 high-quality FL transcripts were obtained through single-molecule real-time (SMRT) sequencing, which were then corrected using Illumina sequencing data. Next, 89,666 nonredundant FL transcripts were generated after mapping to the reference genome of *P. sinensis*; 291 fusion genes and 17,366 novel isoforms were successfully annotated using data from the nonredundant protein sequence database (NR), eukaryotic orthology groups (KOG), the Gene Ontology (GO) project, and the KEGG Orthology (KO) database. Additionally, 19,324 alternative polyadenylation sites, 101,625 alternative splicing events, 12,392 long noncoding RNAs, and 5916 transcription factors were identified. *Smad4*, *Wif1*, and 17-*β-hsd* were identified as female-biased genes, while *Nkd2* and *Prp18* held a higher expression level in males than females. In summary, we found differences between male and female *P. sinensis* individuals in AS, lncRNA, genes, and transcripts, which relate to the Wnt pathway, oocyte meiosis, and the TGF-β pathway. Female-biased genes such as *Smad4*, *Wif1,* and *17-β-hsd* and male-biased genes such as *Nkd2* and *Prp18* played important roles in the sex determination of *P. sinensis*. FL transcripts are a precious resource for characterizing the transcriptome of *P. sinensis*, laying the foundation for further research on the sex-determination mechanisms of *P. sinensis*.

## 1. Introduction

The Chinese soft-shelled turtle (*Pelodiscus sinensis*) is widely distributed in China, Russia, Japan, Thailand, Vietnam, and Korea [1]. In China, it is an economically important aquatic species, and the most commonly consumed turtle species [2]. *P. sinensis* displays significant sexual dimorphism: compared with females, the most obvious characteristics of male individuals are the larger body, faster growth rate, thicker calipash, and lower fat content [3]. Therefore, male *P. sinensis* is associated with a higher market value and breeding benefits. The all-male breeding work was based on the pseudo-female (PF); PF individuals were obtained by a treatment of estradiol (E2) to male embryos in the embryonic development stage, with a male genotype and a female phenotype. And then male individuals were used as the male parent, and PF individuals were used as the female parent for mating and reproduction to produce male offspring.

Sex determination is an important stage in embryonic development, which has a crucial impact on individual morphology and population survival. At present, the mechanisms of animal sex determination can be divided into two categories. One is genotypic sex determination (GSD), and the other is environment-dependent sex determination (ESD); the most typical ESD is temperature-dependent sex determination (TSD), whereas GSD is primarily governed by genetic factors [4]. In the red-eared slider turtle (*Trachemys scripta elegans*), the environmental temperature determined the direction of individual sex differentiation [5], and this situation was also reported in the Asian yellow pond turtle (*Mauremys mutica*) [6,7] and the green sea turtle (*Chelonia mydas*) [8]. In *P. sinensis*, sex determination is performed through genetic sex determination (GSD) [9] and has the same heterologous chromosome system as the spiny softshell turtle (*Apalone spinifera*) [10]. The direction of sex differentiation in GSD species is influenced by the expression of sex-related genes. *Dmrt1*, a classic male development gene, was identified in several species for its role in sexual development regulation [11,12,13]. In *P. sinensis*, *Dmrt1* inhibition results in the retardation of sexual development as well as the male-to-female sexual reversal of gonad development [11]. *Rspo1* is an important regulatory factor during female ontogeny [14]; as such, *Rspo1* deletion or mutation can lead to female-to-male sexual reversal [15]. The loss of function of some genes such as *Foxl2*, *Cyp19a1,* and *Sox9* can impair sexual development [16,17]. Genes not only determine sexual differentiation [18,19,20,21] but also govern growth differences between males and females [22,23].

Gender dimorphism studies have used next-generation transcriptome sequencing technologies with short reads. However, this method has several limitations; for example, reconstructing and quantifying complete transcript isoforms after identifying all the transcript elements provides inaccurate results [24,25]. Single-molecule real-time (SMRT) sequencing, a third-generation sequencing technology created by Pacific Biosciences (PacBio), is widely used in genome sequencing read lengths that can cover different exon connections to obtain full-length (FL) transcripts [26,27,28]. SMRT is a nanopore-based single-molecule reading technology, which can quickly read the base information of each DNA molecule without PCR amplification, avoid the GC bias and coverage heterogeneity caused by amplification, and correct random errors through high coverage. Compared with second-generation sequencing technology, SMRT has a longer sequencing read length, averaging 10–12 kb. The maximum length can exceed 30 kb, and SMRT sequencing has been effectively used for whole-transcriptome profiling in many species [29,30,31]. In addition, gene and transcript annotations based on FL transcripts are more accurate than the assembly of transcripts based on short RNA sequences [32]; moreover, several expressed sequence tags (ESTs) from FL studies of some species aided in improving genome annotation as well as in performing downstream analysis such as expression quantification and alternative splicing (AS), alternative polyadenylation (APA), and long noncoding RNA (lncRNA) identification [33,34,35]. In *Gymnocypris namensis*, an FL transcript sequence was used to generate FL transcripts to explore its adaptability to harsh environments on plateaus; this method resolved limitations such as the inaccurate reconstruction and quantification of complete transcript isoforms after the identification of all transcript elements [36]. The Pacific white shrimp (*Litopenaeus vannamei*) is the first shrimp transcriptome constructed with SMRT sequencing; this method resolves the issues related to the risk of genome misassembly because nearly 80% of the shrimp genome is estimated to comprise repetitive elements. An advantage of SMRT sequencing is that it can provide relatively more comprehensive, detailed genomic information. Compared with traditional sequencing methods, SMRT sequencing can be used to directly read the sequence of a single DNA molecule without fragment splicing, providing more accurate, complete genomic information [37].

Here, we present a transcriptomic analysis conducted using SMRT sequencing; the transcriptome used here was generated from the genes related to sexual maturity in female and male *P. sinensis* individuals. In parallel, Illumina-based paired short RNA reads were used as supporting data for our PacBio-based analyses. The combination of these two technologies may aid in resolving their respective limitations, providing relatively long and accurate transcript data suitable for biological research [38,39]. Our results may enrich the genetic information and further promote the application of transcriptome techniques in *P. sinensis*; moreover, they may pave the way for studies on sex determination and control in Chinese soft-shelled turtles.

## 2. Materials and Methods

### 2.1. Experimental Materials

Three mature females (1000 g; 2-year-old) and three males (1500 g; 2-year-old) of *P. sinensis* were collected from Anhui Xijia Agricultural Development (Bengbu, Anhui Province, China). All turtles were raised in a greenhouse at 30 °C for the first year and in ponds for the second year, which can maintain high growth rate and quality. The gonads of the second-aged turtles were already mature and could lay eggs. Next, 0.3% MS-222 (Sigma-Aldrich, St. Louis, MO, USA), with an active ingredient tricaine methanesulfonate, was mixed with water to anesthetize Chinese soft-shelled turtles (males for 0.5 mg/mL and females for 0.3 mg/mL), which were then dissected. Tissue samples for RNA extraction, including those from the brain, liver, muscle, pituitary, and gonads (ovary or testis), were collected and then stored in liquid nitrogen until RNA extraction. All experimental procedures were performed under the guidelines of the Yangtze River Fisheries Research Institute for the use of laboratory animals.

### 2.2. RNA Extraction

Total RNA was extracted from each sample by using a commercial kit (Takara, Dalian, China), according to the manufacturer’s protocol. After collecting different samples, biological repeated mixing treatments were carried out on the same sex, which could improve the efficiency of the sequencing experiment and reduce the impact of the batch effect on sequencing results. The cell lysate used in this kit was passed through the gDNA Eraser Spin Column and RNA Spin Column to improve the quality of RNA. To eliminate residual genomic DNA contamination, RNA samples were treated with DNase I (Promega, Madison, WI, USA). RNA integrity, quality, and concentration were assessed using an Agilent 2100 Bioanalyzer (Agilent Technologies, Santa Clara, CA, USA). Total RNA samples with RNA integrity number (RIN) ≥ 8 were used for the construction of cDNA libraries for subsequent sequencing experiments.

### 2.3. PacBio Library Construction and Sequencing

Equimolar ratio RNA of different tissues obtained after biological replication was individually pooled for sequence libraries of female and male samples. The FL cDNAs were synthesized using SMARTer PCR cDNA Synthesis Kit (Clontech, Princeton, NJ, USA). Large-scale double-stranded cDNA was generated with an optimal number of polymerase chain reaction (PCR) amplification cycles by using the PCR cycle optimization test. Large-scale PCR products were purified through agarose gel electrophoresis, and the size was assessed using the Bluepippin System (Sage Science, Beverly, MA, USA). After size selection, a second round of PCR was performed to generate a sufficient amount of DNA for SMRT bell template preparation, which was then repurified using the PacBio system. Four SMRT sequencing libraries of size fractionation 0–2 and 2+ kb for both females and males were constructed using size-selected cDNA binding SMRT adapters using Pacific Biosciences DNA Template Prep Kit 2.0 (Sage Science) to create PacBio libraries. Four SMRT cells (Iso-seq PacBio libraries, Menlo Park, CA, USA) were sequenced on the PacBio sequel platform according to its protocol, using P6-C4 reagents with 10 h sequencing movies.

### 2.4. PacBio Data Processing and Error Correction

PacBio raw data were performed using the SMRT Pipeline Analysis software (version 2.3) suite (http://www.pacb.com/products-andservices/analytical-software/smrt-analysis/, accessed on 20 January 2022). In detail, effective reads of inserts (ROIs; as polymerase reads) were obtained from raw sequencing data after being filtered by SMRT Link (version 5.0) with a set of main parameters: minimum subread length, 50 bp; minimum predicted accuracy, 0.8; minimum read score, 0.65; minimum number of full passes, 1. After the deletion of the polymerase read adapter, subread numbers can be calculated. Circular consensus sequence (CCS) reads were generated from the subreads. By identifying the 5′ and 3′ adapters and the poly(A) tail, only the CCS reads with all three signals were considered an FLNC read. FLNC reads after error correction were mapped to the referenced *P. sinensis* genome (IWGSC RefSeq (version 1.0; https://github.com/juliangehring/GMAP-GSNAP, accessed on 2 November 2022). Additional nucleotide errors in FLNC reads were corrected using Illumina RNA-Seq data with Proovread (version 2.12), with the default parameters using parameter coverage of 127.

### 2.5. Gene and Isoform Identification

After error correction and genome comparison, FLNC reads with the same splicing connections were collapsed into one isoform. The isoforms that had a shorter 5′ terminal region but shared introns and splicing sites in the remaining region compared with other isoforms were considered transcripts degraded at the 5′ terminal region and were filtered out. For the remaining isoforms, we examined the supporting evidence. We retained isoforms supported with at least two FLNC reads or with one FLNC read with PID (proportion integral derivative) >99%, or all connection sites were fully supported by Illumina reads or annotations of the RSEM (version 1.3.0). Isoforms overlapped by at least 20% of their length on the same strand were considered to be from the same gene locus. Newly discovered loci and isoforms were identified by comparing identified loci and isoforms with reference genome annotations using the same criteria as those for locus and isoform identification. AS events were classified and characterized by comparing different isoforms of the same gene loci by using Asprofile (version 1.0).

### 2.6. Fusion Transcript Identification

Fusion transcripts were identified by parsing mapping data by using in-house scripts. A FLNC read was considered a candidate fusion transcript when all the following criteria were satisfied: mapping to two or more annotated genes ≥ 10 kb apart, alignment to each gene with >10% FLNC coverage, total combined FLNC coverage from all alignments > 99%, and support by a certain amount of PE reads across the fusion junction.

### 2.7. APA Site Detection

The APA sites for each gene loci were detected using TAPIS (version 1.0) [40]. All FLNC reads aligning to each annotated gene were assessed, and FLNC with 3′ end within 5 bp were grouped using a greedy strategy. A polyadenylation site was considered when it was supported by at least two FLNC reads and was not within 15 bp of a previous polyadenylation site. The numbers of APA sites for each gene locus and those of transcripts supporting an APA site are provided as data files.

### 2.8. LncRNA Identification

To identify lncRNAs from the SMRT sequence data, all isoforms were aligned with sequences from four databases to filter out the probable encoding sequences. The four databases were KOG (https://www.ncbi.nlm.nih.gov/search/all/?term=KOG, accessed on 7 December 2022), clusters of orthologous groups of proteins (COG) (http://www.ncbi.nlm.nih.gov/COG, accessed on 15 December 2022), Swiss-Prot (http://www.expasy.ch/sprot, accessed on 24 March 2023), and NCBI NR (http://www.ncbi.nlm.nih.gov, accessed on 18 April 2023) in NCBI. The sequences with no hits on the aforementioned databases were further screened using CPAT (version 1.2.2) [41]. After filtering the sequences with a coding potential greater than a certain cutoff or length < 200 bp, the remaining transcripts were selected as lncRNA candidates.

### 2.9. RNA-Seq Library Construction and Sequencing

Equal amounts of biologically replicated RNAs from different tissues of female and male *P. sinensis* were pooled. mRNA was isolated from total RNA using NEBNext poly(A) mRNA Magnetic Isolation Module (E7490; NEB, Houston, TX, USA). The cDNA library was constructed using the NEBNext Ultra RNA Library Prep kit for Illumina (E7530, NEB), according to the manufacturer’s instructions. In brief, the enriched mRNA was fragmented into 250–300 bp RNA inserts, which were used to synthesize first- and second-strand cDNA. The double-stranded cDNA was subjected to end-repair/dA-tail and adapter ligation. The suitable fragments were isolated using Agencourt AMPure XP beads (Beckman Coulter, Brea, CA, USA) and enriched through PCR. Library quality was assessed on Agilent Bioanalyzer 2100. Finally, the constructed cDNA libraries of female and male *P. sinensis* were sequenced on HiSeq2500 (Illumina, San Diego, CA, USA).

### 2.10. RNA-Seq Data Analysis

Low-quality reads, such as only adapter, unknown nucleotides > 5%, or containing more than 50% of low-quality (Q-value ≤ 20) bases, were removed using a custom-made Perl script. The filtered clean reads of female and male individuals were separately mapped to the reference genome sequence (PelSin 1.0; https://www.ncbi.nlm.nih.gov/assembly/GCF000230535.1, accessed on 20 January 2022) of *P. sinensis* using Tophat2 (version 2.1.1) [42]. The aligned records from the aligners in the BAM/SAM format were further examined to remove potential duplicate molecules. Gene expression levels were estimated using fragments per kilobase of exon per million fragments mapped (FPKM) values using Cufflinks (version 2.1.1) [43]. DESeq2 (version 1.16.1) [44] and Q-value were used to evaluate differential gene expression between female and male *P. sinensis* individuals. For each transcription region, an FPKM (fragment per kilobase of transcript per million mapped reads) value was calculated to quantify its expression abundance and variations using RSEM software (version 1.3.1) [45]. The false discovery rate (FDR) control method was used to identify the threshold *P* value in multiple tests to compute the significant difference. Only genes with an absolute value of log_2_ (fold change) of ≥2 and FDR of <0.01 were used for subsequent analysis.

For more accurate annotation of obtained transcripts, all of them were blasted against the different databases using BLASTx (http://www.ncbi.nlm.nih.gov/BLAST/, version 2.2.24) to evaluate sequence similarity with genes of other species. The following databases were included: NCBI NR, Swiss-Prot, COG, and KOG. GO annotation was analyzed using Blast2GO (version 2.2), and KEGG pathway mapping was performed using Kobas (version 3.0).

### 2.11. DEG Validation through RT-qPCR

The accuracy of the transcriptome data was validated through RT-qPCR on eight DEGs. Primers were designed using Primer Premier (version 7.0; Appendix A). β-Actin was the internal control. Total RNA was extracted from the gonads of mature males and females of *P. sinensis* by using Trizol (Invitrogen, Carlsbad, CA, USA). First-strand cDNA template was synthesized from total RNA by using the PrimeScript RT Reagent Kit (Takara, Dalian, China). PCR was performed on a 7500 Real-Time PCR System using SYBR Select Master Mix (2X) (Applied Biosystems, Carlsbad, CA, USA). All experiments were performed in triplicate. The 2^−ΔΔCt^ method was used to calculate gene relative expression. The data were analyzed using one-way analysis of variance (ANOVA), followed by Duncan’s test in SPSS (version 22.0; IBM, Amonk, NY, USA). *p* < 0.05 was considered to indicate significance.

## 3. Results

### 3.1. SMRT Sequence Analysis

We used the PacBio Iso-Seq platform to perform SMRT sequencing to identify as many *P. sinensis* transcripts as possible (Appendix A). Total RNA was extracted from five tissues: brain, pituitary, liver, muscle, and ovary or testis; they were pooled and examined before library construction. Four libraries of two pooled size-fractionated (M/F < 2 kb and M/F > 2 kb) poly(A) RNA samples from the two sexes of *P. sinensis* were constructed, sequenced, and barcoded using SMRT sequencing.

From these four libraries, a dataset with 43,442,083,322 bp and about 2,780,628 polymerase reads was generated. After sequences < 50 bp in length were filtered out, the cleaned read numbers of the four libraries were 42,550, 33,750, 36,750, and 32,750 (Table 1). After filtration and correction, 42,585,573,542 filtered subread data with a mean length of 1683 bp (<2 kb), 2708 bp (>2 kb), 1571 bp (<2 kb), and 2422 bp (>2 kb) for four libraries were obtained. In total, 1,536,849 circular consensus sequence (CCS) reads with a mean depth of 7–14 passes were obtained from all the sequences of four SMRT cells after filtering with SMRT Link (version 4.0) for each of the four libraries (Appendix A). CCS reads were classified into five types: with a 5′ adapter, with a 3′ adapter, with a poly(A) tail, FL, and FL nonchimeric (FLNC). The FLNC reads were simultaneously nonchimeric sequence reads with a complete poly(A) tail as well as 5′ and 3′ barcoded primers. Here, the FLNC read number was 1,121,532, representing 72.97% of all CCS reads; the mean FLNC read length was 1572–3124 bp (Table 2, Appendix A).

### 3.2. Functional Annotation of Transcripts

Female and male *P. sinensis* individuals demonstrated considerable differences in body size (Figure 1A). To further understand this difference, we obtained 89,666 isoforms and 30,670 unannotated loci including loci <1 k 2096 (6.83%), loci 1–2 k 8975 (29.26%), loci 2–3 k 8682 (28.13%), and loci >3 k 10,971 (35.77%; Appendix A) by the redundant transcript removal and false-positive filtration of the loci and isoform. By analyzing the transcript length distributions, we found that our PacBio dataset retrieved much longer transcripts than those described in the reference annotations (Figure 1B). Of the 89,666 isoforms, 17,366 from 15,172 novel loci were identified as novel isoforms. To validate the unannotated novel isoforms, we searched the 17,366 novel isoforms in the NCBO Nonredundant (NR), gene ontology (GO), KO, and eukaryotic ortholog groups (KOG) databases, and 344 novel isoforms with remarkable hits in the four databases were found (Figure 1C). The functions of all FL transcripts were predicted by the KOG database, and 1069 transcripts were divided into 26 KOG classifications (Figure 1D). The largest component of the categories was general function prediction only (*n* = 202, 18.90%), followed by signal transduction mechanisms (*n* = 180, 16.84%); post-translational modification, protein turnover, and chaperones (*n* = 139, 13%); and the cytoskeleton (*n* = 85, 7.95%).

### 3.3. Analysis of LncRNAs and AS Events

We further evaluated the coding potential of sequences without a hit in the NR, KOG, and KO databases by using the coding potential assessment tool (CPAT). After filtering coding potential greater than a certain cutoff or a length of <200 bp, we used the remaining sequence as a result of the final noncoding RNA prediction. In total, 12,392 lncRNAs were eventually detected, representing 16.56% of all novel isoforms. The average length of these lncRNAs was 1973 bp, with most lncRNAs ranging in length from 500 to 3500 bp (Figure 2A). LncRNAs were classified into five groups: lncRNAs generated from the intergenic regions (51.33%), intronic regions (25.68%), sense strand (14.44%), and antisense strand (5.17%) and other lncRNAs (3.38%). In addition, the mean length of lncRNAs in males was 2023 bp, whereas, in females, it was 1608 bp (Figure 2B), suggesting that lncRNAs play different roles between male and female *P. sinensis* individuals.

Among the AS events detected by the data, if there were differences in the splicing sites marked by yellow dots, it is considered that this AS event has approximate boundaries. Adding “X” before the abbreviation of this AS event represented the diversity of the splicing sites in this AS (Appendix A). Within the constructed transcriptome, we detected 101,625 AS events, including 22,174 exon skipping (SKIP), 7520 multiple cassette exon skipping (MSKIP), 5066 single intron retentions (IR), 1654 multiple intron retentions (MIR), 8147 alternative exon ends (AE), 18,564 approximate exon skipping (XSKIP) (“X” was used for 19,034 approximate multiple cassette exon skipping (XMSKIP), 8332 approximate single intron retentions (XIR), 858 approximate multiple intron retentions (XMIR), and 10,276 approximate alternative exon ends (XAE). The AS events demonstrated considerable sex differences: among all AS events, SKIP demonstrated the highest frequency in both females (42.97%) and males (21.52%), whereas XMIR exhibited the lowest frequency in females (0.84%) and males (0.95%; Figure 2C). In total, 647 and 51,060 AS events were detected in females and males, respectively (Figure 2D).

### 3.4. APA Analysis

SMRT sequencing has been a powerful tool for plant APA research; however, it has not been widely used to analyze *P. sinensis* transcriptomes. Multiple mRNA isoforms with different coding sequences or 3′ untranslated regions can be produced through APA. APA plays an essential role in determining the transcriptome complexity, thereby potentially regulating the function, stability, localization, and translation efficiency of target RNAs. Here, we created a precise genome-wide map of different APA sites based on FL isoforms, with 19,324 APA sites identified from 10,700 genes, and 4329 had more than one APA site among these genes (Appendix A). Thus, APA sites are widely distributed in the *P. sinensis* transcriptome.

### 3.5. Analysis of Novel Genes and Isoforms in Males and Females

In the present study, the number of genes expressed in males was more than that expressed in females (Figure 3A). In both females and males, there were more upregulated genes and isoforms, 5050 and 11,058, respectively (Figure 3B). In particular, 40.01% and 30.07% of known genes were upregulated and downregulated, respectively, whereas 18.44% and 11.48% of novel genes were upregulated and downregulated, respectively (Figure 3C). Similarly, 20.49% and 13.74% of known isoforms were upregulated and downregulated, respectively, and 39.66% and 26.11% of novel isoforms were upregulated and downregulated, respectively (Figure 3D). In addition, we identified 291 fusion genes related to 1907 genes; notably, the fusion fragments of all the fusion genes originated from different chromosomes.

To further evaluate the sex differences in *P. sinensis*, we performed cluster analysis on the novel genes of males and females. All the novel genes were classified into three types, whereas all the novel isoforms were classified into four types; the first type of genes included 2054 genes, whereas the second type included 4281 genes (Figure 4A,B). The gene and isoform expression trends for the second type were consistent in both females and males, whereas their third and fourth types demonstrated an opposite trend in males (Figure 4C,D).

### 3.6. KEGG Enrichment Analysis of Genes and Isoforms

To assess the biological function of the *P. sinensis* transcriptome overall, the FL transcripts were further annotated by mapping the sequences into reference canonical pathways in KEGG. In total, 174 pathways with 72 upregulated genes in the Wnt pathway (ko04310), oocyte meiosis (ko04114), and the TGF-β pathway (ko04350) were found (Figure 5A). In total, 134 pathways with 127 downregulated genes involved in ovarian steroidogenesis (ko04913), glycerophospholipid metabolism (ko00564), and peroxisome (ko04146; Figure 5B) were noted. Regarding isoforms, meiosis (ko04113), ovarian steroidogenesis (ko04913), and α-linolenic acid metabolism (ko00592) shared 111 upregulated differentially expressed genes (DEGs), accounting for 52, 32, and 27, respectively (Figure 5C). Finally, 278 DEGs related to endocytosis (ko04144; *n* = 173), the p53 pathway (ko041150; *n* = 42), and the cell cycle (ko04110; *n* = 63) were significantly downregulated (Figure 5D).

### 3.7. Validation of DEGs through Quantitative Real-Time Polymerase Chain Reaction

Eight sex-related DEGs were randomly selected for quantitative real-time polymerase chain reaction (RT-qPCR) validation (Table 3). *Smad4*, a part of the TGF-β pathway, is involved in cell proliferation and differentiation; we observed considerable differences in *Smad4* expression levels between males and females. We noted that *Wif1* expression was higher in females than in males, possibly because *Wif1* expression is related to embryonic development. The expression levels of *Cypj2* and *17β-hsd* were significantly higher in females than in males, whereas *Nkd2* and *Prp18* were significantly higher in males than in females (Figure 6).

## 4. Discussion

Sexual dimorphism is widespread throughout the animal kingdom [46]. Understanding the dynamics of gene expression can aid in extensively uncovering sex fate choice and sexual dimorphism establishment [47]. Short-read sequencing on the Illumina platform facilitates the effective detection of gene expression and AS events. However, its capacity to accurately detect FL splice variants of genes is limited. The long-read lengths of the PacBio system are well-suited for characterizing FL cDNA isoforms produced from high-quality poly(A) RNA; however, they cannot provide information regarding isoform expression and structures. Here, we used Pacific Biosciences (PacBio)’s isoform sequencing to identify alternative splicing events and more novel isoforms to improve genome annotation. Meanwhile, more lncRNAs were identified, and fusion genes were accurately obtained, which lays a foundation for the subsequent study of the molecular mechanism of *P. sinensis*.

In this study, we obtained 1,121,532 high-quality FLNC reads, with 89,666 isoforms covering 30,670 loci, 101,625 AS events corresponding to 10 types, 291 fusion genes from inter-chromosomes, and 19,324 poly(A) sites from 10,700 genes and 12,392 lncRNAs (accounting for 16.56% of the novel isoforms). This new resource and transcriptional information may improve the genome annotation of livestock transcriptome research on *P. sinensis*.

Our experimental design aimed at maximizing transcript diversity and investigating comprehensive splice isoforms by broadly sampling different tissues from males and females. The average length and read N50 length of FL transcripts were noted to reach 2400 and 38,000 bp, respectively, in females and 1997 and 34,750 bp, respectively, in males—much longer than those of Illumina RNA-seq transcripts. The presence of longer isoforms indicated higher sequence integrity among the species, which is crucial for subsequent gene function and evolutionary studies. As novel >200-nucleotide-long non-protein-coding RNAs, lncRNAs play important roles in many biological and pathological processes, such as immune responses, cell-cycle control, splicing, differentiation, and epigenetic regulation [48]. However, no lncRNA in *P. sinensis* has been previously reported. Here, we identified 12,392 lncRNA transcriptomes; the mean lncRNA length significantly differed between males and females [49,50]. The pre-mRNA splicing process involves intron removal and protein-coding element assembly into mature mRNAs. Alternative pre-mRNA splicing, a major source of transcriptome and proteome complexity, involves selectively joining different coding elements to form mRNAs that encode proteins with functions similar to or distinct from the proteins encoded by the original mRNAs [51,52,53].

AS is essential for regulating molecular, cellular, physiological, and developmental processes and pathways in eukaryotes [54]. Here, we identified 649 and 51,060 AS events in male and female individuals, respectively—indicating that AS events demonstrate considerable sex differences in *P. sinensis*. As an important precursor RNA processing mechanism [55], APA finely regulates gene expression, and alterations in poly(A) sites represent a major type of RNA post-transcriptional regulatory modification [56]. We identified 19,324 poly(A) sites derived from 10,700 genes; of them, 4329 genes had multiple poly(A) sites, and the rest demonstrated a single poly(A) site. APA affects mRNA stability, translation efficiency, and other important processes through poly(A) site selection. Two hundred and ninety-one fusion genes from inter-chromosomes were identified. Fusion genes can lead to some genetic changes with significant effects [57]. These genetic changes may involve the regulation of growth hormones, cell division and differentiation, organ development, and metabolic processes [58]. These changes may directly affect an individual’s growth rate, height, weight, and other characteristics.

We validated our SMRT data using eight DEGs. First, *Smad4*, the central mediator of TGF-β signaling [59] with important roles in many biological processes, demonstrated differences in expression levels between females and males. *Wif1*, an inhibitory factor in the Wnt pathway, demonstrated an expression trend identical to that of *Smad4*. Because *Smad4* can activate the Wnt pathway [60], whether it has antagonistic effects with *Wif1* [61] or facilitates the growth process in *P. sinensis* warrants research. *Wdr24* is a key effector of *Gator2*, a subcomplex of *Torc1*, which is a major eukaryotic metabolism regulator [62]. However, we noted no significant difference in *Wdr24* expression between males and females. The effects of *Wdr24* expression on metabolism may be a reason for the differences between male and female individuals. Moreover, *Wdr24* expression is associated with the expression of genes including *Dmrt1*, *Sox9*, and *Foxl2* in *P. sinensis*. In the process of studying the sex determination mechanism of GSD species, it is necessary to pay attention to the expression changes in these two genes and their related pathways to further study the mechanism of sex differentiation. The study of genes associated with faster and greater growth in males is indeed an interesting approach that could potentially provide insights into the molecular mechanisms underlying sexual dimorphism in growth rates. One potential avenue of investigation is through the study of genes involved in hormonal regulation [63]. Previous research demonstrated that sex hormones, such as estrogen and testosterone, play a key role in regulating growth and development in mammals [64]. Estrogen influences development by transmitting its signals through estrogen receptors, including nuclear receptors α and β and the membrane receptor GPER. However, the sequential relationship between the estrogen receptor and the growth hormone/insulin-like growth factor 1 (GH/IGF-1) axis in ontogeny is worthy of further study. This result provides a new research direction for our next study on growth and sex differentiation mechanisms in *P. sinensis*.

In general, our study demonstrated that long-read sequencing complemented by short-read sequencing affords the in-depth cataloging and quantification of eukaryotic transcripts. Based on the FLNC transcripts, our results revealed AS events, the role of fusion genes, and the difference in lncRNAs and DEGs between males and females. The results screened out significant sex-related signaling pathways and genes. The female-biased genes such as Smad4, Wif1, and 17-β-hsd and the male-biased genes such as Nkd2 and Prp18 can be a marker of females and males, which can help to identify the sex to assist the next step of breeding in the industry. These data are also helpful to further elucidate the molecular mechanism of sex differentiation in *P. sinensis*. Our results not only significantly improve existing gene models of *P. sinensis* but also provide crucial rules and generate novel resources and information with positive implications for further research on the molecular mechanisms underlying sex regulation in *P. sinensis*.

## 5. Conclusions

In summary, we used SMRT sequencing and RNA-seq to obtain the comprehensive FL transcriptome of *P. sinensis*. The obtained FL transcript isoforms are accurate, reliable, and, thus, applicable for gene annotation, molecular marker development, and lncRNA prediction. After identifying the female-biased genes (*Smad4*, *Wif1*, *Cypj2*, and *17β-hsd*) and male-biased genes (*Nkd2* and *Prp18*), we found that *Wdr24* is associated with many genes related to sex differentiation such as *Dmrt1*, *Foxl2*, *Cyp19a1* and *Sox9*. Moreover, we also found 291 fusion genes, so we will further investigate whether the growth difference between male and female individuals is related to those fusion genes. Our results may facilitate future comparative studies on the transcriptome and gene function of *P. sinensis*.

## Figures and Tables

**Figure 1 animals-13-03704-f001:**
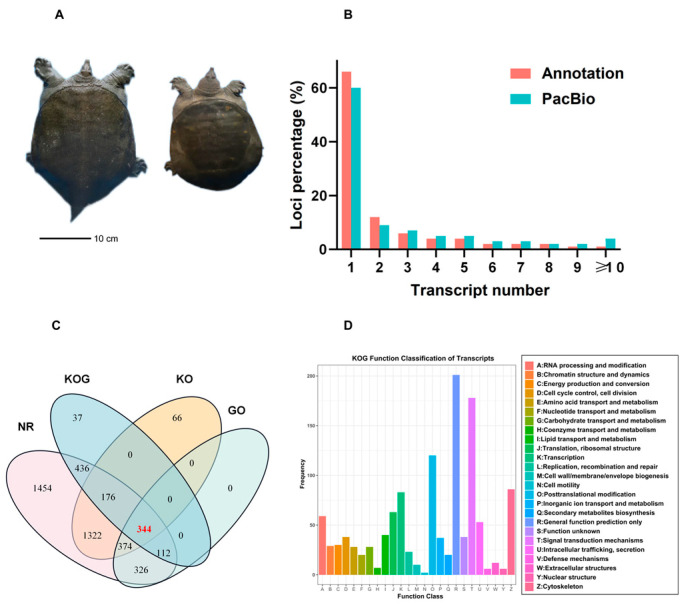
Experimental results. (**A**) Male and female *P. sinensis*. (**B**) Isoform distributions within the same loci. (**C**) Novel genes identified in SMRT sequencing data. The numbers represent novel genes identified in the NR, KOG, KO, and GO databases. (**D**) KOG analysis of isoforms.

**Figure 2 animals-13-03704-f002:**
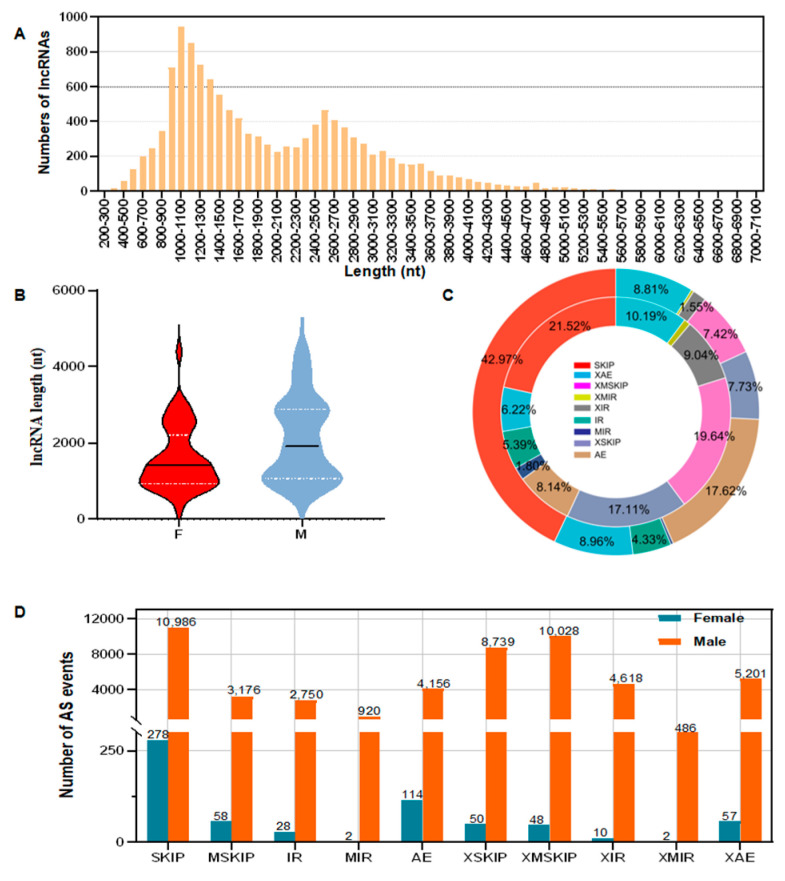
Analyses of lncRNAs and AS events. (**A**) Distribution of lncRNAs in length ranges. (**B**) Average lengths of lncRNAs in males and females. The bold black line represents the lncRNA length at the median value. The white dotted lines above and below represent the lengths of the lncRNAs at the upper quartile (FPKM–UQ) and lower quartile (FPKM–DQ), respectively. (**C**) Proportion of AS events in males and females. The outer and inner rings represent females and males, respectively. (**D**) Numbers of AS events in males and females. If the two splicing sites marked by yellow dots are not the same, they are considered approximate boundaries (Approximate); “X” is added before abbreviations of alternative splicing events.

**Figure 3 animals-13-03704-f003:**
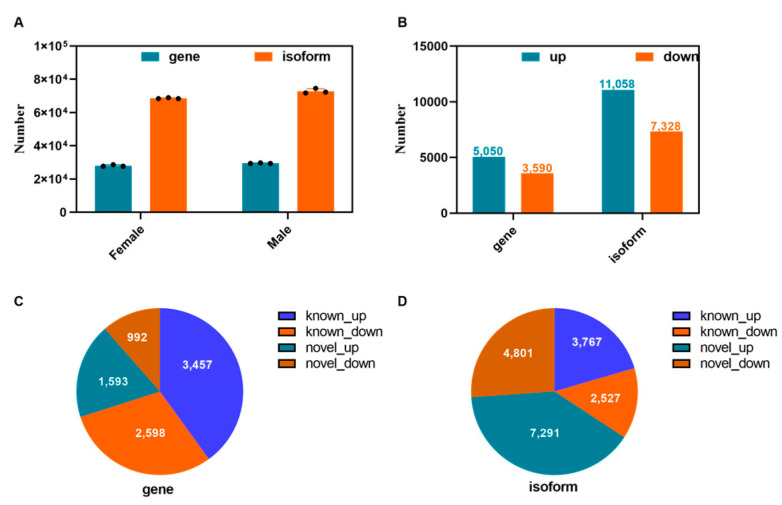
Gene and isoform in four *P. sinensis* libraries. (**A**) Numbers of expressed genes and isoforms. The black dots represent 3 different samples in each group. (**B**) FPKM box plot of genes. (**C**) Proportions of different gene types. (**D**) Proportions of different isoform types.

**Figure 4 animals-13-03704-f004:**
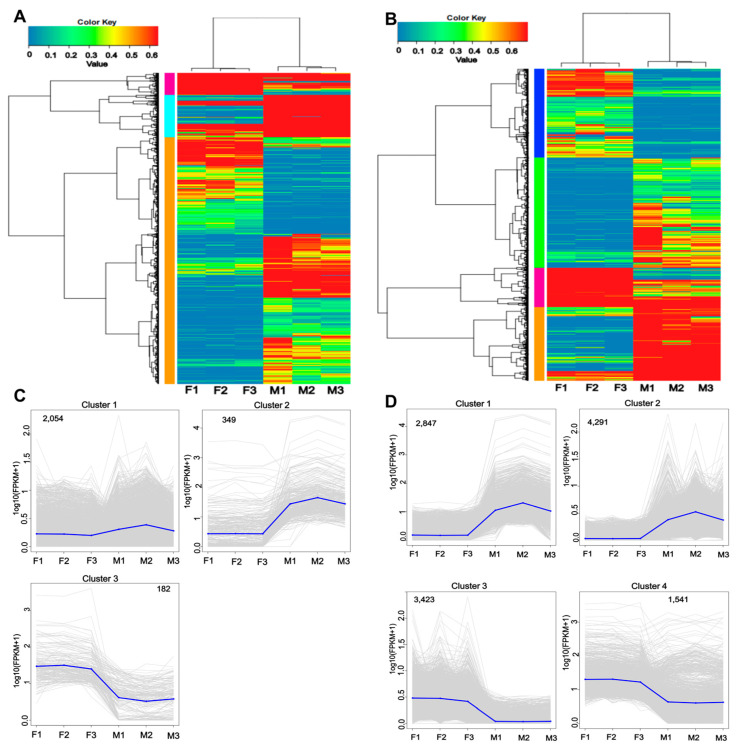
Analysis of differential genes and isoforms between males and females. (**A**) Cluster analysis of differential genes between males and females. (**B**) Cluster analysis of differential isoforms between males and females. (**C**) The log_10_ (FPKM + 1) line diagram of differential genes. (**D**) The log_10_ (FPKM + 1) line diagram of differential isoforms.

**Figure 5 animals-13-03704-f005:**
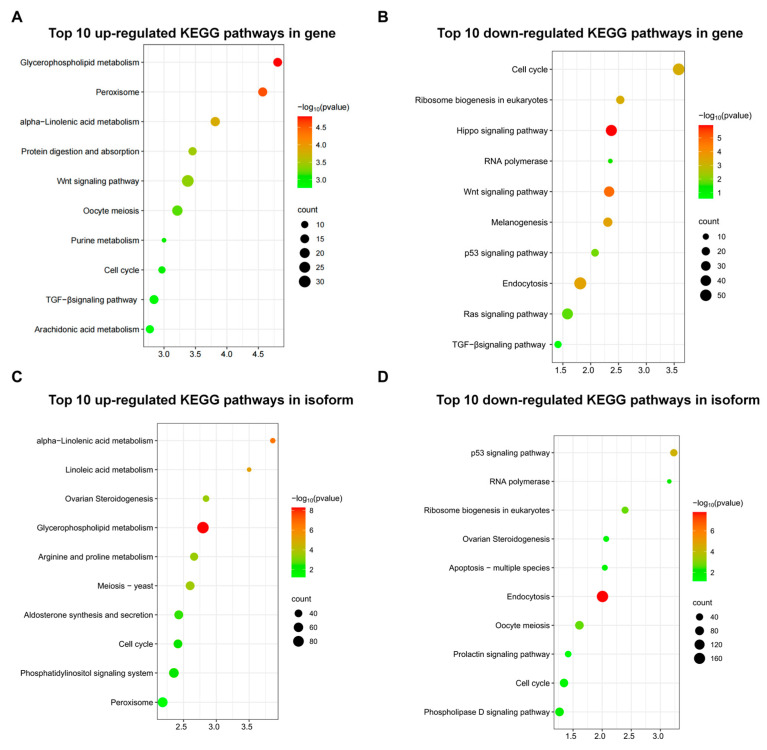
Comparison of significantly enriched KEGG terms of DEGs among the groups. (**A**) Top 10 upregulated KEGG pathway genes. (**B**) Top 10 downregulated KEGG pathway genes. (**C**) Top 10 upregulated KEGG pathway isoforms. (**D**) Top 10 downregulated KEGG pathway isoforms.

**Figure 6 animals-13-03704-f006:**
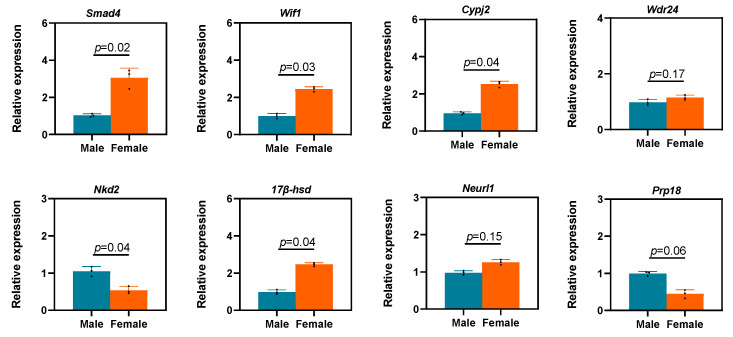
Validation of SMRT sequencing data through RT-qPCR. The expression levels of differential expressed genes such as *Smad4, Wif1, Cypj2, Wdr24, Nkd2, 17β-hsd, Neurl1 and Prp18* between males and females. Each value is presented as the mean ± SD of three repetitions. One-way ANOVA with Tukey post-hoc tests were used to analyze the means.

**Table 1 animals-13-03704-t001:** PacBio libraries and sequencing results.

Library	Total Bases (bp)	ROIs	Mean Length (bp)	Read N50
F 0–2 k	12,461,189,105	661,271	18,844	42,250
F 2 k+	10,011,411,220	686,536	14,582	33,750
M 0–2 k	10,292,266,219	680,809	15,118	36,750
M 2 k+	10,677,216,778	752,012	14,198	32,750
Total	43,442,083,322	2,780,628	-	-

F and M indicate female and male *P. sinensis*, respectively; “0–2 k” indicates that the length of the library was less than 2 kb; “2 k+” indicates that the length of the library was greater than 2 kb.

**Table 2 animals-13-03704-t002:** Classification analysis of CCS reads.

Library	CCS	5′ Primer	3′ Primer	Poly(A)	FL	Of FLNC	Mean FLNC Length (bp)
F 0–2 k	427,494	386,518	387,826	367,398	347,788	327,713	1675
F 2 k+	337,154	286,996	288,005	269,671	244,810	239,600	3124
M 0–2 k	400,518	352,792	356,931	333,916	312,660	296,777	1572
M 2 k+	371,683	304,514	310,103	288,588	259,044	257,442	2856
-	1,536,849	1,330,820	1,342,865	1,259,573	1,164,302	1,121,532	-

F and M indicate female and male *P. sinensis*, respectively. “0–2 k” indicates that the length of the library was less than 2 kb; “2 k+” indicates that the length of the library was greater than 2 kb.

**Table 3 animals-13-03704-t003:** Candidate differentially expressed genes.

Gene	Description	Log_2_ (Fold Change) (F vs. M)
*Smad4*	Smad family 4	2.33
*Wif1*	Wnt inhibitory factor 1	2.73
*Nkd2*	Naked cuticle 2	−3.25
*17β-hsd*	17-β-Hydroxysteroid dehydrogenase	−6.22
*Cypj2*	Cytochrome P450, subfamily J, family 2	2.11
*Neurl1*	Neuralized E3 ubiquitin protein ligase 1	−3.25
*Prp18*	Proline-rich protein 18-like	−2.46
*Wdr24*	Wd repeat domain 24	2.22

F and M indicate female and male *P. sinensis*, respectively.

## Data Availability

The datasets presented in this study can be found in online repositories. The names of the repository/repositories and accession number(s) can be found below: Genome Sequence Archive, accession number: CRA013253.

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
