# Peer review of "Single-Molecule Real-Time Sequencing for Identifying Sexual-Dimorphism-Related Transcriptomes and Genes in the Chinese Soft-Shelled Turtle (Pelodiscus sinensis)"

_animals, 2023, doi:10.3390/ani13233704_

Round 1

Reviewer 1 Report

Comments and Suggestions for Authors

The study conducted by the researchers is a comprehensive analysis of the transcriptome of the Chinese soft-shelled turtle (Pelodiscus sinensis) using PacBio isoform sequencing. The study identified 1,536,849 high-quality full-length transcripts, which were then corrected using Illumina sequencing data. After mapping to the P. sinensis reference genome, the researchers generated 89,666 non-redundant full-length transcripts. The study also identified 291 fusion genes, 17,366 new isoforms, 19,324 alternative polyadenylation sites, 101,625 alternative splicing events, 12,392 long non-coding RNAs and 5,916 transcription factors. The study found that genes such as Wif1 and Wdr24 are associated with the WNT signaling pathway and sex-related genes such as Dmrt1, Foxl2Cyp19a1 and Sox9. The study concludes that full-length transcripts are a valuable resource for characterizing the transcriptome of P. sinensis and for understanding the role of genes in gonadal development and sex determination.

The present study yielded a considerable amount of genomic information. However, its main limitation is its novelty, and the results do not provide conclusive evidence for the role of sex-related genes such as Dmrt1, Foxl2, Cyp19a1 and Sox9 in the mechanism of sexual differentiation and the observed growth patterns in this species.

In addition, there are several issues that need to be addressed to improve the manuscript:

Introduction

-          The introduction is unbalanced in terms of the importance of its main themes. There is a short, general paragraph on the turtle specie and importance of their exploration. This is followed by a rather weak paragraph on GSD and finally two slightly more detailed paragraphs discussing the benefits of single-molecule real-time (SMRT). The introduction could be greatly improved by including the existing literature on chelonids and sexual differentiation.

Methodology

-          PACBIO's SMRT sequencing methodology provides a robust approach for generating high quality data. The importance of this method lies in its ability to provide accurate and comprehensive genomic information. Please explain why this method is better than others.

Results

-          Was tissue pooling performed? And why? The interesting aspect is the observation of whether certain genes are expressed more or less. However, when pooling is performed, only genes are detected without being assigned to a specific tissue and/or sex.

Discussion

-          Female turtles exhibit accelerated growth compared to their male counterparts, and the study of genes associated with faster and greater growth in females is an interesting approach. Let’s address this aspect in the discussion and compare it with the existing literature.

-          What conclusions can be drawn from the introduction about the role of genes such as Dmrt and Rspo1 in Pelodiscus sinensis?

-       It is questionable to use only β-Actin for normalization of gene expression without providing information on the stability of this reference gene, which is known to fluctuate in some cases. The use of only one reference gene for normalization of gene expression quantification is not in accordance with current standards. 

Conclusions

-          There is a lot of genomic data obtained in the study that could be discussed, why only a few are mentioned in the conclusion? I suggest working more with the data obtained and comparing it with the literature to draw a more detailed conclusion. Also mention future work or research directions that can be developed based on the results obtained in this study

Reviewer 2 Report

Comments and Suggestions for Authors

Journal: Animals (ISSN 2076-2615)
Manuscript ID: animals-2720136
Type: Article
Title: Single-molecule real-time sequencing for identifying sexual dimorphism-related transcriptomes and genes in Pelodiscus sinensis
Section: Aquatic Animals
Special Issue: Mechanisms of Sex Determination and Reproduction in Aquatic Animals

In this research, the authors used SMRT sequencing to obtain the transcriptome from males and females of the commercial species Pelodiscus sinensis, to study expression differential patterns between sexes and, in the following research, the sex determination of this species.  It should be noted that from a commercial point of view, males would, in principle, be of greater interest due to their higher growth rate, among other reasons. In my opinion, there is a lot of effort, as shown by the many results obtained. Nevertheless, I strongly suggest clarifying the objectives of this research; more information is required in the Introduction and Discussion section accordingly, as I have indicated below. I have issues and questions about the Material and Methods section as well. For me, the figures should be separated into bigger figures for the main text and Supplementary data. After the Introduction section, the Material and Methods section must be placed. Sorry, but I could not find Supplementary Tables S1-S3 anywhere.

Comments on the Quality of English Language

Minor suggestion of English

Reviewer 3 Report

Comments and Suggestions for Authors

Animals-2720136-peer-review-Report-1

The manuscript explores Single-molecule real-time sequencing to identify sexual dimorphism-related transcriptomes and genes in Chinese soft-shelled turtles, Pelodiscus sinensis. The authors analyse AS, lncRNA, genes, and transcript differences between male and female P. sinensis individuals. Their findings could enhance genetic information, advance proteomic techniques in this species, and pave the way for sex determination and control studies in soft-shelled turtles.

The manuscript is well-written and backed by current citations. However, I have noted areas for improvement to enhance its acceptability for publication and these are presented below.

The title of the manuscript is okay but could also be ‘Single-molecule real-time sequencing for identifying sexual dimorphism-related transcriptomes and genes in the Chinese soft-shelled turtle (Pelodiscus sinensis)’. This is just a suggestion that could be adopted or ignored.

Line 89: the word ‘to’ is missing between ‘sequencing’ and ‘identify’.

Lines 248-251: This sentence appears more of reporting the results rather than discussion. It should be paraphrased to flow with the next sentence.

Lines 280-281: Citation needed.

Lines 303-309: It would be good for the authors to state the number of male and female turtles used in the study.

Lines 304-305: The concentration of the anaesthetic agent was stated, and how much of it was administered per turtle (Dosage per animal, considering the difference in weight between males and females)? It is essential that the active ingredient of the anaesthetic agent is mentioned as well.

Lines 311-312: The authors stated that the total RNA was extracted from each sample by using a commercial kit (Takara, Dalian, China), according to the manufacturer’s protocol. It will be worthwhile for this protocol to be stated here in a few sentences for their readers to follow and understand.

Lines 430-436: Conclusion. It should provide a summary of your findings, highlighting key results and their significance. The authors should acknowledge any limitations in the study, suggest directions for future research and emphasize the overall contribution of the study. The conclusion should, therefore, be paraphrased to take into account the points raised above.

Lines 458-569 References: One of the high points of this manuscript is that most of the citations used in writing the paper are from current journals. However, the authors did not follow the MDPI guidelines in writing these references. For instance, where there are many authors in a citation, the authors shorten the other authors by writing etc, instead of et al. Most of the journal titles were written in upper cases, and those with more than two names were not abbreviated. Reference number one should be Gong, S.; Vamberger, M.; Auer, M.; Praschag, P.; Fritz, U. Millennium-old farm breeding of Chinese softshell turtles (Pelodiscus spp.) results in massive erosion of biodiversity. The Science of Nature. 2018, 105, 1-0.

The Animals MDPI format/ sequence was not strictly followed. It should be as follows: Title, abstract, introduction, materials and methods, results, discussion, conclusions…….and references. The authors should bring materials and methods before the results.

Comments on the Quality of English Language

These have been pointed out in the attached report.

Reviewer 4 Report

Comments and Suggestions for Authors

The manuscript titled "Single-molecule real-time sequencing for identifying sexual dimorphism-related transcriptomes and genes in Pelodiscus sinensis" is a well-performed study on the transcriptome and gene expression regulating sex determination and gonadal development in the turtle Pelodiscus sinensis. The authors utilized the SMRT Sequence Analysis method, which is appropriate for the stated objectives. Reptiles are a poorly explored group in terms of gene expression controlling sexual development, and they have unknown mechanisms of sex determination. The paper should be accepted for publication after making minor revisions such as improving punctuation, commas, and spaces between words. In line 22 and 123, it should be clarified whether it is "euKaryotic" or "eukaryotic." In line 165, the sentence "the sentence ‘X’ is addend before…..” is not clear. Abbreviations need to be expanded upon their first usage. Figure 1D is entirely unreadable. The font size on Figures 1, 2, 4, and 5 is too small, making the labels on the figures completely illegible. This must be corrected.

Round 2

Reviewer 1 Report

Comments and Suggestions for Authors

The new version of the manuscript submitted by the authors has adequately addressed all of my inquiries.

Author Response

Thanks for your comments. 

Reviewer 2 Report

Comments and Suggestions for Authors

The authors have responded adequately to the comments raised during the review process. It should be noted that data availability will not be open until October 30, 2025 (see: https://ngdc.cncb.ac.cn/search/?dbId=gsa&q=CRA013253). Sorry, but I did not understand this time moratorium that does not facilitate the review process but would not allow the scientific community to access the data even while reading/citing this manuscript. Due to the Open Access philosophy of the MDPI publisher this must be changed:

"Deposition of Sequences and Expression Data

New sequence information must be deposited to the appropriate database PRIOR to submission of the manuscript. Accession numbers provided by the database should be included in the submitted manuscript. MANUSCRIPTS WILL NOT PUBLISHED until the accession number is provided."
Source: https://www.mdpi.com/journal/animals/instructions

I add some minor modifications.

Line 116: It would be better higher sample size. All samples were under the same conditions (e.g., temperature)? Please indicate it.

Line 218: I strongly recommend changing "the Perl script" to "a custom-made Perl script".

Figure S1. "Quality filtering" is two times. Add a "g" at the end of "Alternative splicin".

Figure S4. Maybe in plural form? "Number of genes" instead of "number of gene" and "ployes" instead of "ploy". Please review it, carefully.

Comments on the Quality of English Language

Minor edition of language was required.

Reviewer 3 Report

Comments and Suggestions for Authors

Comments on the Quality of English Language

The quality of the English language used in writing the paper is high; minor alterations and errors have been pointed out to the authors and are stated in the report.
